# Influence of Ionizing Radiation on Spontaneously Formed Aggregates in Proteins or Enzymes Solutions

**DOI:** 10.3390/pharmaceutics15051367

**Published:** 2023-04-29

**Authors:** Karolina Radomska, Marian Wolszczak

**Affiliations:** Institute of Applied Radiation Chemistry, Faculty of Chemistry, Lodz University of Technology, 93-590 Lodz, Poland

**Keywords:** nanoparticle, nanotechnology, pulse radiolysis, proteins, dityrosine, aggregation, fluorescence, oxidation

## Abstract

We have shown that many proteins and enzymes (ovalbumin, β-lactoglobulin, lysozyme, insulin, histone, papain) undergo concentration-dependent reversible aggregation as a result of the interaction of the studied biomolecules. Moreover, irradiation of those protein or enzyme solutions under oxidative stress conditions results in the formation of stable soluble protein aggregates. We assume that protein dimers are mainly formed. A pulse radiolysis study has been made to investigate the early stages of protein oxidation by $N_{3}^{•}$ or ^•^OH radicals. Reactions of the $N_{3}^{•}$ radical with the studied proteins lead to the generation of aggregates stabilized by covalent bonds between tyrosine residues. The high reactivity of the ^•^OH with amino acids contained within proteins is responsible for the formation of various covalent bonds (including C–C or C–O–C) between adjacent protein molecules. In the analysis of the formation of protein aggregates, intramolecular electron transfer from the tyrosine moiety to Trp^•^ radical should be taken into account. Steady-state spectroscopic measurements with a detection of emission and absorbance, together with measurements of the dynamic scattering of laser light, made it possible to characterize the obtained aggregates. The identification of protein nanostructures generated by ionizing radiation using spectroscopic methods is difficult due to the spontaneous formation of protein aggregates before irradiation. The commonly used fluorescence detection of dityrosyl cross-linking (DT) as a marker of protein modification under the influence of ionizing radiation requires modification in the case of the tested objects. A precise photochemical lifetime measurement of the excited states of radiation-generated aggregates is useful in characterizing their structure. Resonance light scattering (RLS) has proven to be an extremely sensitive and useful technique to detect protein aggregates.

## 1. Introduction

Protein nanostructures, due to their unique properties and small size, have aroused great interest among researchers in recent years. Protein-based nanostructures can be used as therapeutic agents applied for biomedical research. Albumin nanoparticles have many applications, including drug delivery systems (DDS) in anticancer therapy or cosmetology [1]. The protein-based nanoparticles have demonstrated to be a promising alternative for a new generation of drug delivery systems, in particular for chemotherapeutical or oncotic agents. Proteins and enzymes, e.g., bovine serum albumin and papain, have been studied at the nanoscale [2,3] and have demonstrated high potential for drug delivery systems due to their small size and high biological affinity. Albumin aggregates used as DDS are characterized by properties that improve biocompatibility, distribution and physicochemical properties of the drug in a living organism. On the other hand, the aggregation of protein drugs is of significant concern. Many recently approved protein pharmaceuticals, as well as those in clinical trials, are manufactured and stored in aqueous solutions. Referring to our earlier work and the literature date, it can be concluded that proteins and enzymes undergo concentration-dependent reversible aggregation in neat solutions.

The influence of oxidative stress on protein structure has been described in detail in the literature. Papers describing reactions of albumins with radicals generated under conditions of reduction stress are much rarer. Radicals and reactive oxygen species interacting with proteins induce the formation of albumin aggregates stabilized with intermolecular bonds. New bonds formed between biomolecules depend mainly on the nature of the radicals generated in the solution (oxidizing or reducing radicals). A dityrosine bridge is observed in many oxidatively modified proteins, e.g., human or bovine serum albumin, ovalbumin, insulin and lysozyme. The formation of intermolecular covalent bridges, including S–S, C–C or C–S bonds, is also observed by the irradiation of protein and enzyme solutions. During protein aggregation in neat protein solutions, conformational changes in their structure can be registered, and they consist mainly in increasing the content of the β-sheet structure. Hydrogen bond networks with β-sheet cross-structure and/or π–π stacking interactions in protein aggregates are the cause of specific emission in albumin solutions [4]. Radiation-induced aggregates show specific emission different from non-irradiated solutions (emission from self-aggregates). Spontaneous aggregation occurs in some polypeptide solutions, and light-excited aggregates exhibit visible emission spectra. The new “luminescence” is not related to the presence of aromatic amino acids in albumin, but to electron delocalization through the network of amino acids of neighboring HSA molecules. Such structures are stabilized by hydrogen bonds, but hydrophobic interactions, electrostatic forces and van der Waals forces are also most likely involved in the formation of new chromophore groups. Unlike the major HSA chromophore groups, the structure of these newly formed ones is not exactly known. Based on the results of our measurements for a number of albumins, proteins and enzymes, we believe that there are many different groups of amino acids responsible for the “blue” emission.

An increased amount of dimeric and oligomeric forms of albumin is often the first symptom of the development of many diseases (e.g., type II diabetes, Alzheimer’s or Parkinson’s disease, etc.). All these diseases are accompanied by the formation of insoluble fibrous aggregates [5]. The key factor in albumin aggregation is the process of dityrosyl cross-linking, which is the result of the recombination of tyrosyl radicals in adjacent protein molecules. From a medical point of view, it is necessary to propose an effective method to identify pathogenic amyloids. Understanding the mechanism of protein aggregation is fundamental for their identification. In order to understand and control pathological albumin aggregation, it is necessary to distinguish between the different processes involved in aggregation and to identify which are the most relevant. Several techniques can be used to analyze protein aggregates, including size exclusion chromatography, gel electrophoresis, dynamic light scattering, mass spectrometry, NMR spectroscopy or turbidity measurements. A very useful measurement technique for studying the process of protein aggregation is FTIR (Fourier transform infrared spectroscopy), which enables the tracking of individual phases of the discussed process. However, there is no single effective analytical method for every protein. For this reason, we became interested in spectroscopic techniques to identify protein aggregates with light emission detection. In this work, we focused on well-known proteins and enzymes. The main motivation of our work is the basic science that contributes to understanding the mechanism of the formation of protein aggregates for future applications in selective drug delivery.

## 2. Materials and Methods

### 2.1. Sample Preparation

Proteins and enzymes were obtained from Sigma-Aldrich (St. Louis, MO, USA)/ Merck (Darmstadt, Germany) and were used as received. All solutions were prepared in ultrapure water in phosphate buffer 10 mM pH 7.2. Water was purified with the Hydrolab SPRING 20UV system. Proteins and enzymes were dissolved in PBS (phosphate-buffered saline) solution to appropriate concentrations immediately before measurements. Final concentrations of the solutions were verified spectrophotometrically using the molar absorption coefficients: ε_280nm_ = 35,600 M^−1^ cm^−1^ for ovalbumin (Sigma-Aldrich, A5503), ε_280nm_ = 53,500 M^−1^ cm^−1^ for papain (Roth, 8933.1), ε_280nm_ = 18,000 M^−1^ cm^−1^ for β-lactoglobulin (Sigma-Aldrich, L3908), ε_280nm_ = 39,000 M^−1^ cm^−1^ for lysozyme (Sigma-Aldrich, 62970), and ε_280nm_ = 5800 M^−1^ cm^−1^ for insulin (Sigma-Aldrich, 91077C). The histone from calf thymus was obtained from Sigma-Aldrich (H9250).

### 2.2. Optical Measurement

Absorption spectra were recorded with the resolution of 0.5 nm using a Perkin Elmer Lambda 750 spectrophotometer. Steady-state emission and excitation spectra of examined solutions were carried out using Aminco-Bowman Series 2, equipped with a xenon lamp and a red-sensitive photomultiplier (Hamamatsu R928, Hamamatsu Photonics, Shizuoka, Japan). Typically, the voltage of the photomultiplier divider was 650 V. The excitation and emission slits were set to 4.0 and 2.0 nm, respectively. Measurements of absorption and emission spectra were made for single scans, but often verified in additional measurements. The samples were prepared immediately before measurements by dissolving tested proteins in PBS solution with appropriate concentration (concentrations of the solutions were verified spectrophotometrically). To measure the emission intensity of spontaneous OVA aggregates as a function of albumin concentration, a stock solution containing 180 µM was diluted to the desired concentrations.

The fluorescence and emission spectra presented in the publication did not require correction of the instrument response and/or the effect of the internal filter. All emission spectra of protein solutions were verified (an example is shown in Appendix A in Appendix A). In the present work, we compared uncorrected and corrected spectra of β-lactoglobulin (700 μM) after oxidation with azide radicals (λexc = 320 or 337 nm, see Appendix A in Appendix A). In the case of β-lactoglobulin aggregates, after excitation of the protein solution with light of 337 nm wavelength, the use of emission correction shifts the spectrum by 4 nm (412 nm → 415 nm) and 6 nm (418 nm → 424 nm) for the neat and irradiated solution, respectively. In this study, we focused on the qualitative characteristics of proteins and enzyme dimers and aggregates; therefore, it is not necessary to correct the emission intensity value.

### 2.3. Pulse Radiolysis

Pulse radiolysis was performed using a 6 MeV linear accelerator (LINAC) ELU-6E operating in a single pulse mode. The equipment for pulse radiolysis with the optical detection has been described elsewhere [6]. The flow system was used in the pulse radiolysis of examined solutions with the volume of 250 mL. The solutions were N_2_O saturated during irradiation. Water radiolysis and the process of formation of reactive radical species used to initiate radical reactions have been described in detail in the literature [7]. In the pulse radiolysis measurements, we used two measurement regimes. In the nanosecond domain, we most often used a pulse with a duration of 17 ns generating 55 Gy. The dose was determined based on data for hydrated electron [8] or thiocyanate as a dosimeter. In measurements with microsecond pulses, we used pulses with a duration of 1 or 4 μs. The typical dose for a 1 μs pulse is 200 Gy, and for a 4 μs pulse, it is approximately 800 Gy. In these cases, we applied an alanine dosimeter to measure the dose. Pulse radiolysis measurements were carried out under anaerobic conditions to avoid subsequent reactions of the formed radicals with oxygen.

### 2.4. Dynamic Light Scattering

The size of generated aggregates of proteins was monitored by dynamic light scattering on a Zetasizer Nano ZS90 device (Malvern Instruments Ltd., Malvern, Worcestershire, UK). Size measurements were made with a buffer solution of β-lactoglobulin (70 μM) or papain (70 μM) before and after irradiation in the presence and absence of NaN_3_ (0.1 M). Polymethacrylate cuvettes with an optical path length of 1 cm were used for DLS measurements. The volume of the tested solution was always 1.5 mL. Before measurement, the samples were filtered using 0.45 μm cellulose acetate syringe. Nanoparticle size was determined at 25°C using backscatter angle (173°) in triplicates of 3 runs of 18 each, using program to protein analysis. The average size of protein aggregates was determined on the basis of number-based sizes analysis.

## 3. Results

### 3.1. Radiolysis of Protein Solutions

To better understand the mechanism of protein aggregation induced by ^•^OH or $N_{3}^{•}$ radicals, we conducted pulse radiolysis experiments and spectroscopic measurements for solutions of peptide and proteins, including papain, ovalbumin, insulin, calf thymus histone, β-lactoglobulin or lysozyme. The HSA oxidation process related to electron transfer has been investigated by pulse radiolysis and photochemical methods and has recently been described by us [4]. The aim of our study was an attempt to answer the question of whether it would be possible to observe the formation of DT both by pulse radiolysis and spectroscopic techniques. In living organisms, there are two major reactive oxygen species (ROS), superoxide and hydroxyl radicals that are being continuously formed in a process of the reduction in oxygen to water. The consequence of an excess of ROS in organisms, especially ^•^OH radicals, is many diseases, such as atherosclerosis, cancer and neurological disorders. The hydroxyl radical is a very reactive species. For this reason, the reactions of ^•^OH radicals with various compounds are not selective. The primary mechanisms of the reactions of an ^•^OH radical with organic compounds involve OH addition and H abstraction and, to lesser extent, electron transfer reactions (preferentially on the surface of albumin molecules). The azide radical is one of the most important one-electron oxidants commonly used in radiation chemistry studies involving molecules of biological significance. In contrast to the ^•^OH radical, $N_{3}^{•}$ radical appears to react primarily via one-electron transfer and it is much more selective in its reaction than is the hydroxyl radical. However, there are several reports indicating the possibility of azide radical addition to organic compounds containing double bonds in aqueous solutions [9,10]. One of the methods of protein cross-linking is the formation of dityrosine bridges, which are formed as a result of the recombination of long-lived tyrosyl radicals (TyrO^•^) generated under oxidative conditions. Our own study [4] and data presented in the literature [11,12] indicate that in aqueous solutions of tyrosine, TyrO^•^ radicals are formed as one of the products of reaction with hydroxyl radicals. As a result of the reaction of ^•^OH or $N_{3}^{•}$ radicals with HSA molecules, stable protein aggregates of low and high molecular weight are formed. Covalent bonds are detected between biomolecules, and at low radiation doses, mainly dimers are observed. In the case of mild oxidants as azide radicals, HSA aggregates are characterized by classic DT fluorescence. Our measurements clearly revealed that $N_{3}^{•}$ penetrates to the HSA interior [4]. However, in the reaction of albumin with ^•^OH radicals, essentially, we did not observe DT. The high reactivity of ^•^OH radicals with protein amino acids results in a greater probability of reaction with non-aromatic protein residues, and thus a lower efficiency of dityrosine bridge formation. For this reason, in our experiments, we also used the azide radical to study the process of one-electron oxidation of proteins.

Many reactions important from a medical point of view of proteins involve charge transfer, including those initiated by ionizing radiation. Kinetic experiments have conclusively shown that electron transfer from a tyrosine residue (donor of electron) to the tryptophanyl radical (acceptor of electron) can take place over large distances through protein interiors. The tryptophan radical (reduction potential about 1V at pH = 7) within protein shell oxidizes tyrosine residue to the phenoxyl radical (TyrO^•^). Both radicals absorb light in different spectral regions, which is an advantage in radiolysis measurements to track electron transfer. The long-range intramolecular electron transfer (LRET) process is much more complicated than was previously thought. For example, the LRET in the hen egg-white lysozyme accompanying radical transformation Trp^•^ → TyrO^•^ was observed [13,14,15]. However, some tryptophan and tyrosine residues are not involved in the long-range intramolecular electron transfer process, or LRET does not always take place in the protein structure. In the case of lysozyme and β-lactoglobulin, our pulse radiolysis results provide helpful information with regard to electron transfer in protein solutions. In both cases, electron transfer accompanying radical transformation Trp^•^ → TyrO^•^ was observed. This process is easy to observe for relatively low molecular weight proteins (~20 kDa). When $N_{3}^{•}$ radicals react with proteins containing Trp and Tyr residue, one-electron oxidation of the TrpH residue is followed by an efficient, rapid intramolecular process, which could be regarded as hydrogen atom transfer from TyrOH to Trp^•^. In the case of a reaction of azide radicals with lysozyme (Figure 1) or β-lactoglobulin (see Appendix A in Appendix A), kinetic analysis indicates that the decay of Trp^•^ is correlated with the formation of TyrO^•^ (the trace at 410 nm evolves with the same kinetics as Trp^•^ decays). The decay of Trp^•^ and formation of TyrO^•^ within β-lactoglobulin structure takes place on a time scale of microseconds. For β-lactoglobulin, which has been studied in detail, the first-order transformation Trp^•^ → TyrO^•^ is independent of protein concentration, showing that the charge transfer is intramolecular rather than intermolecular. Our measurements are in agreement with the literature [16,17,18,19] (Figure 1A,C of Ref. [17]).

Our pulse radiolysis experiments confirmed that the reaction mechanism of azide radicals with lysozyme is significantly different from the processes associated with the attack of hydroxyl radicals on this protein. The attack of ^•^OH radicals on lysozyme mainly induces the formation of tyrosyl radicals (Figure 2). When lysozyme reacts with $N_{3}^{•}$ radical, the absorption spectrum shows a distinct band in the spectral range of 450–600 nm. The analysis of pulse radiolysis measurements clearly indicates that the main reaction product of the $N_{3}^{•}$ radical with lysozyme are the Trp^•^ radicals. However, the concentration of TyrO^•^ radicals in lysozyme (absorption band with a maximum at 410 nm) was lower compared to other pulsed irradiated proteins [13,20,21]. Electron transfer over long distances (though not as effective as in the case of lysozyme) was also observed in irradiated HSA solutions [22]. The authors estimated that in the case of HSA, the yield of charge transfer Trp^•^ → TyrO^•^ is about 20%. The disappearance of the Trp^•^ radical concentration (determined in radiolysis measurements while taking into account the molar Trp^•^ coefficient: 1750 dm^3^ mol^−1^ cm^−1^ at 520 nm) and the increase in the TyrO^•^ radical concentration (molar absorbance coefficient 2700 dm^3^ mol^−1^ cm^−1^ at 410 nm) in the case of HSA are not identical. The publication [22] showed, by measuring the maximum concentration of the TyrO^•^ radical (buildup absorbance at 410 nm), that the efficiency of electron transfer over long distances is only 20%. Analysis of the kinetics of Trp^•^ radical decay shows that almost 75% of these radicals have disappeared over the time scale of TyrO^•^ formation. The disappearance of the TyrO^•^ radical within seconds has been linked to the formation of DT. The presence of one tryptophan group in the HSA structure makes this reaction channel less important in our case.

When papain or OVA reacts with $N_{3}^{•}$ radicals (Figure 3), similar transient absorption spectra were obtained as in the case of the reaction of the azide radical with HSA [4]. Pulse radiolysis measurements of the N_2_O-saturated buffer solutions (pH 7.2; 0.1 M NaN_3_) of all studied proteins and the enzyme were performed (in the separated experiments). The reaction of $N_{3}^{•}$ with HSA, OVA and papain [23] leads to the formation of the Tyr meta isomer (band centered at ~305 nm), phenoxyl radicals TyrO^•^ (band with a maximum near 410 nm) and Tyr^•^ radicals (band with a maximum at 520 nm). In the spectral range 300–400 nm, the light Trp^•^ and cation radical TrpH^•+^ are also absorbed. In contrast to the abovementioned proteins, the main process of the reaction of $N_{3}^{•}$ radicals with insulin is the formation of TyrO^•^ (band with a maximum at 410 nm). Significant differences were observed in the 300–400 nm region, where most of the aromatic radicals absorb light [10]. Both Trp^•^ and its proton adduct also absorb in this region. For this reason, changes in spectral range 300–400 nm may result from different degrees of protonation of aromatic radicals. The pH effects on the transient absorption spectra of Trp in the presence of azide radicals suggest that several transients contribute to the observed absorption spectra in this region [10].

Recently, we have shown that in the reaction of azide radicals with BSA, the intensity of the absorption band originating from Trp residues is twice as intense as that recorded for the HSA solution. This indicates that both tryptophan residues in BSA are accessible to azide radicals in a similar way. After normalizing the spectra obtained in the initial stage of oxidation of the examined objects (Figure 3), three main bands (with maxima 305, 410 and 520 nm) are observed, but the spectra are not identical. The correlation of the shape of the spectrum of the primary oxidized samples with the ratio of the number of tryptophan and tyrosine residues is clearly observed (see Appendix A in Appendix A). The intensity of the band (spectrum normalized to the radical band of the tryptophan residue) associated with the Tyr^•^ radical is the lowest for lysozyme (smallest tyrosine residue/tryptophan residue ratio: 0.5 (3/6)). For papain and OVA, the pulse radiolysis transient spectra are similar and the Tyr/Trp ratio values are similar: 3.8 for papain (19/5) and 3.33 (10/3) for OVA.

The literature data show that ovalbumin agglomerates are linked by covalent bonds [24]. Pulse radiolysis measurements indicate that OVA aggregation is not a cross-linking process. There is no recombination of OVA radicals formed as a result of the reaction with hydroxyl radicals. Figure 4 presents the transient absorption spectra of primary products of OVA one-electron oxidation by azide radicals (N_2_O-saturated 60 µM OVA solution containing 0.1 M NaN_3_). These spectra are compared with the spectra obtained after oxidation of OVA by ^•^OH radicals (N_2_O-saturated 60 µM OVA solution). The transient absorption spectrum of the oxidation products of OVA by ^•^OH radicals can be considered structureless except for the maximum at 300 nm, shoulder at 350 and 410 nm and weak absorbance at 530 nm [24]. The absorption spectra of the protein after the reaction with ^•^OH radicals were difficult to analyze due to the lack of selectivity of the hydroxyl radical, which resulted in the formation of many different radicals. The hydroxyl radical can react with all the amino acids in proteins (preferentially on the protein surface), but the azide radical reacts mostly with Tyr and Trp. For this reason, the OVA spectrum after reaction with hydroxyl radicals is less resolved than for the azide radical. The insert in Figure 4 shows transient absorption spectra recorded at various times after the electron pulse irradiation (200 Gy) of N_2_O-saturated aqueous solution containing 220 µM OVA and 0.1 M NaN_3_.

In the case of ovalbumin, we do not observe the Trp^•^ → TyrO^•^ transformation (we did not observe a build-up of absorption at 410 nm). The decay of absorbance at 410 and 510 nm within OVA structure takes place on a time scale of milliseconds in contrast to HSA, where this process takes place on a time scale of seconds. In the reaction of papain with azide radicals, charge transfer between tryptophan and tyrosine was also not observed.

Figure 5 shows transient absorption spectra recorded at various times after the electron pulse irradiation of N_2_O-saturated aqueous solution 75 µM papain and 0.1 M NaN_3_. The insert in Figure 5 shows the kinetics pattern of transient absorption decay recorded for 415 and 525 nm for both Trp^•^ and TyrO^•^ radical decays according to complex kinetics. The absence of an increase in absorbance due to the oxidation of TyrOH to TyrO^•^ occurring simultaneously with the reduction in Trp^•^ to Trp may indicate a lack of electron transfer from the tyrosine residue to the tryptophanyl radical in the papain structure. Such intramolecular electron transfer is effective, for example, in β-lactoglobulins (see Appendix A in Appendix A), but in the case of albumins, it is ineffective and slow, taking place in the seconds. In some cases, the charge transfer is essentially stoichiometric or does not take place in the protein structure. The complex kinetics of the decay of both bands associated with TyrO^•^ and Trp^•^ radicals and the large measurement noise do not allow an attempt to extract information about a possible partial long-distance transfer from the decay of the 410 nm band in our case. For this reason, we do not exclude a transfer Trp^•^ → TyrO^•^ in the case of papain.

Histone is a protein with a relatively low molecular weight (less than 23 kDa) that is characterized by a high content of basic amino acids, mainly lysine and arginine. There are only two cysteine residues within a histone structure; no cystine and tryptophan residues are in the polypeptide chain. It is known from the literature that the rate constant of the reaction of histones with ^•^OH radicals change significantly with the ionic strength and pH of the solution, in contrast to BSA, where no influence of the ionic strength on the reaction with the hydroxyl radical was found [25]. An acidic environment inhibits the process of histone aggregation, and high pH values favor this process. We conducted pulse radiolysis experiments for His in a buffer solution of pH 7.2, because below pH 6, the aggregated form of His is unstable.

Appendix A (see in the Appendix A) shows transient absorption spectra recorded at various times after 17 ns electron pulse irradiation (dose 55 Gy) of N_2_O-saturated aqueous solutions containing 1.4 mg/mL histone and 0.1 M NaN_3_. The selected absorption spectra indicate that the reactions of the used oxidizing species (^•^OH or $N_{3}^{•}$) with the histone lead to the formation of a limited number of products. In the case of azide radicals, new absorption in the visible region with a maximum near 400 nm is due to the phenoxyl radical of tyrosine, TyrO^•^. Our study confirms that hydroxyl radicals are not effective (essentially, there is no signal of TyrO^•^, see Appendix A in Appendix A) in generating TyrO^•^ radicals and, consequently, DT formation in histone solutions, which is consistent with previous studies by other researchers. The transient absorption spectrum of N_2_O-saturated buffer solutions containing histone is in agreement with the absorption spectrum of histone solution after irradiation described in the literature [25,26]. Histones do not contain tryptophan residues in their structure and have a smaller number of aromatic amino acids compared to human or bovine albumin; therefore, the transient absorption spectrum recorded in pulse radiolysis is not intense, as it was for HSA or OVA.

We are aware that the accessibility of radicals to individual amino acids in the polypeptide chain of various proteins is important in the analysis of the albumin oxidation process. Pulse radiolysis measurements combined with enzyme activity tests allowed to determine which amino acid residues in biomolecules are modified as a result of reactions with free radicals [27,28].

### 3.2. Spectroscopic Analysis of Protein Aggregation Process Induced by Ionizing Radiation

In this paragraph, we focused on the photochemical characterization of protein dimers and aggregates generated by irradiation under oxidative conditions (^•^OH or $N_{3}^{•}$ radical). Our DLS measurements suggest that many proteins tend to dimerize prior to irradiation. The formation of protein dimers is accompanied by the appearance of a new “blue” emission in the spectrum. Fortunately, the luminescence of light by radiation-induced protein aggregates differs from the emission of self-aggregates (mainly dimers) [4]. In most cases, as a result of the irradiation of protein solutions during oxidative stress, the formation of dimers/aggregates leads to an emission band with a maximum of about 400 nm. In the same spectral region, a band characteristic of dityrosine is observed. Steady-state pulse radiolysis measurements can be divided into two groups: (a) measurements for protein solutions with a concentration below 100 µM and (b) measurements for concentrated protein solutions (above 300 µM). The vast majority of measurements were made for solutions with a low albumin concentration (low albumin concentration allows minimizing emissions from high-molecular aggregates). The consequence of the formation of protein dimers in neat solutions is a difficult analysis of the size of aggregates generated by ionizing radiation. As it turned out, as a result of irradiation of protein solutions, in most cases, spontaneous (non-covalent) dimers are replaced with dimers stabilized by a cross-link. Moreover, the presence of dimers in neat protein solutions make it difficult to distinguish spontaneous aggregates from those generated by radiation which apply the DT identification procedure proposed by Davies et al. [20]. This is due to the fact that the peak of the DT fluorescence band is in the same region in relation to the peak of the emission band of the spontaneous dimers or aggregates. The difficulties described above will be shown on the example of experiments carried out for lactoglobulin solutions. This protein molecule contains two tryptophan and 4 tyrosine residues. Excitation of N_2_O-saturated and irradiated aqueous β-lactoglobulin (70 μM) solution containing NaN_3_ (0.1 M) with the 320 nm light leads to the emission corresponding to the emission of DT centered at near 400 nm, as shown in Figure 6A. This indicates the presence of dimers stabilized by a covalent bond between the tyrosine residues after irradiation. Excitation of irradiated β-lactoglobulin solution (Figure 6A) with light above 320 nm does not lead to the appearance of new emission bands in the spectrum of protein. The band observed above 370 nm after excitation with a 295 nm light of irradiated N_2_O-saturated β–lactoglobulin solution indicates energy transfer from *Trp214 to protein aggregates (FRET—Förster resonance energy transfer) (Figure 6). This is evidenced by the detection of the emission of both ^*^Trp and protein aggregates after the excitation of the irradiated β-lactoglobulin solution with 295 nm light absorbed only by Trp residue (red line, Figure 6A). In the case of irradiated histone solution (in the case of ^•^OH and $N_{3}^{•}$ radicals) or lysozyme ($N_{3}^{•}$) under oxidative conditions, energy transfer was less efficient. Tyrosine molecule excited in neutral aqueous solutions shows the fluorescence of both the acidic and alkaline form of Tyr (shoulder around 295 nm as shown in Figure 6B) [4].

When β-lactoglobulin reacts with ^•^OH radicals, species other than DT are formed. Two populations of protein aggregates are observed after irradiation of N_2_O-saturated β-lactoglobulin solution (70 µM, 5600 Gy). The strong scattering of light due to aggregate formation is particularly well-visible in the case of concentrated solutions of β-lactoglobulin after reaction with the ^•^OH radical (see Appendix A in Appendix A).

Analysis of the dynamic light scattering measurement (DLS) after irradiation of N_2_O-saturated solution containing β-lactoglobulin (70 μM) confirmed the formation of protein aggregates. In the native solution, β-lactoglobulin forms dimers with mean particles of about 4 nm in size. [29]. It is important to note that β-lactoglobulin exists as a stable dimer (36.7 kDa) at pH values near its isoelectric point (i.e., pH 5.2) at room temperature [30,31,32]. The formation of dimers, trimers and higher oligomers was reported to be more extensive at pH 7.5. The presence of dimers in a neat solution before irradiation can significantly affect the process of protein aggregation. Moreover, β-lactoglobulin solution irradiated in a lower protein concentration (3 mg/mL) showed more aggregates than in higher concentration at the same radiation dose [32,33]. After irradiation of the N_2_O-saturated β-lactoglobulin solution with dose 5600 Gy, two populations of protein aggregates are observed (size of generated nanoparticles was about 20 nm and 52 nm). The reaction of the azide radical with β-lactoglobulin leads to the formation of mainly protein dimers (the size of generated species was around 8 nm).

The important observation from DLS measurements was the presence of high molecular aggregates in the N_2_O-saturated and irradiated β-lactoglobulin solution. The ^•^OH radical can react with all the amino acids in the protein structure (preferentially on the protein surface), but $N_{3}^{•}$ reacts mostly with Tyr and Trp. The high reactivity of hydroxyl radicals with protein amino acids results in a lower relative yield of oxidation of aromatic residues and a lower probability of DT formation. In the case of non-selective, highly reactive species such as hydroxyl radicals, there is a high probability that a TyrO^•^ radical from one oxidized β-lactoglobulin molecule will react with a non-TyrO^•^ radical formed on another modified albumin molecule. The $N_{3}^{•}$ radical leads to the formation of only radicals located on aromatic amino acid residues, mainly on or near the surface of proteins. Thus, the probability of the recombination of TyrO^•^ radicals generated on the surfaces of two different protein molecules increases, which leads to the formation of intermolecular DT.

The time-resolved fluorescence measurements for β-lactoglobulin solution before and after irradiation revealed that there is an increase in the life-time of β-lactoglobulin aggregates generated by ionizing radiation. Decay at 410 nm is characterized by the fluorescence lifetime 3.55 ns before irradiation and the lifetime is about 3.76 ns after irradiation of β-lactoglobulin solution (see Appendix A in Appendix A). The determined emission lifetime values made it possible to distinguish self-aggregates of β-lactoglobulin from aggregates induced by ionizing radiation on the basis of time-resolved fluorescence measurements. In our opinion, the use of a more accurate measurement technique based on single photon-counting technique should help distinguish protein aggregates obtained by the oxidation of albumin molecules with azide or hydroxyl radicals.

The emission spectra of a saturated N_2_O aqueous solution containing 70 µM lysozyme and 0.1 M NaN_3_ were recorded after irradiation with dose 3200 Gy and shown in Figure 7. The specific DT fluorescence band dominates in the lysozyme spectrum. A similar conclusion regarding the formation of DT was reached in the case of irradiation of lysozyme under conditions in which the ${\mathrm{Br}}_{2}^{•-}$ radicals were the oxidizing agent [21,34]. Franzini et al. reported that lysozyme aggregates (28 and 42 kDa) appeared above the dose of 20 Gy absorbed during radiolysis, and their amount was correlated with the increase in DT emission, which suggests the formation of a covalent bond between tyrosine residues in the lysozyme structure [35]. In our experiments, the excitation of the lysozyme solution with light that does not excite endogenous luminophores (λ_exc_ > 380 nm) leads to emission in the spectral range 460–560 nm. Lysozyme is an example of a protein for which, in addition to the formation of covalent bonds as a result of the attack of $N_{3}^{•}$ radicals, aggregates emitting light in the visible range have been observed.

Figure 7B shows the excitation spectra recorded for the emission wavelength at 410 or 450 nm for a lysozyme after reaction with $N_{3}^{•}$ radicals. The band centered at about 330 nm for the detection wavelength of 410 nm is typical for dityrosine. DT bridging is accompanied with the formation of light-emitting aggregates in the low-energy part of the spectrum (an emission band extending over 375 nm, maximum band at ≈3.1 eV). The similar excitation spectra were obtained for the N_2_O-saturated solution containing 70 μM HSA and 0.1 M NaN_3_ after irradiation [4] (Figure 18 of Ref. [4]).

The molecular basis of insulin aggregation is relevant for modeling the amyloidogenesis process, which is involved in many pathologies [36]. Insulin is known to form polymorphic amyloid fibrils depending on aggregating conditions. It was shown that the appearance of the intrinsic fluorescence of insulin fibrils is strongly dependent on the protonation state of N- and C-termini in the fibrils [37]. Photochemical analysis of the measurement data obtained for saturated N_2_O and irradiated (1485 Gy) insulin solution (140 μM) containing NaN_3_ (0.1 M) shows that DT and new intermolecular hydrogen and/or ionic bonds were formed in the solution (Figure 8A). The excitation of the solution with ~320 nm light leads to a wide emission band of insulin aggregates (suggest two populations of aggregates). What is important to note is that insulin does not contain any tryptophan residues in polypeptide chains. The scavenging of ^•^OH radical by insulin also leads to species other than DT, as evidenced by a broad, less intense red-shifted band (λ_max_ = 440–450 nm, λ_exc_ = 370–390 nm, Figure 8B). In addition, the results of near and far UV-CD spectroscopy described in the literature [38] show that exposure of insulin to UV also leads to structural damage, including insulin dimerization by cross-linking dityrosine or breaking the disulfide bond. When insulin reacts with ^•^OH or $N_{3}^{•}$ radicals, in both cases, two populations of aggregates were observed. The insulin (51-residue peptide) monomer consists of two polypeptide chains linked by two disulfide bonds. For this reason, we can expect that certain aromatic groups of insulin will be more readily available for attack by the radical in contrast to more complex systems as proteins. It is reported that at least one tyrosine residue must be accessible to ^•^OH radicals in the monomer of insulin [39]. In the case of the insulin molecule, $N_{3}^{•}$ can also react with accessible cysteine, although the reaction is about 100 times slower than its reactions with Trp and Tyr. Recently, amyloid aggregation has been studied at neutral pH, which more closely resembles the physiological conditions [37]. A very interesting observation was the detection of the fluorescence spectrum of the insulin solution after excitation with 350 nm light, with maximum centered at 440 nm. The authors proposed that intrinsic blue-green fluorescence can be used to study different fibrillization pathways. Moreover, this intrinsic fluorescence may offer an alternative method for the detection of the amyloid formation without using external probes.

Recently, our studies on albumin aggregation suggest that the appearance of visible fluorescence does not require the presence of aromatic residues or a conjugated π-electron system. The same conclusion concerning the origins of intrinsic fluorescence of insulin amyloids was drawn by Iannuzzi et al. [37]. Radiolysis of insulin under oxidative conditions leads to the formation of aggregates characterized by a new emission of light. However, there is a significant difference in the position of the emission bands of insulin after reaction with hydroxyl or azide radicals. This is probably due to the different mechanism of $N_{3}^{•}$ and ^•^OH radicals with insulin. The excitation spectrum recorded after the reaction of the $N_{3}^{•}$ radical with insulin, with wavelength 350 nm, is similar to that of the obtained for insulin amyloids [37]. In our opinion, redox processes associated with insulin oxidation leads to the formation of aggregates whose emission spectrum corresponds to that of insulin amyloids. Further, our work is well underway to clarify whether the spectral convergence is to some extent due to the involvement of covalent bonds in the formation of insulin amyloids.

Ovalbumin is the major protein component of egg white. The OVA polypeptide chain consists of 385 amino acid residues (44.3 kDa). The emission spectra of neat OVA solutions in the concentration range of 60–180 µM has been registered to study the self-aggregation process of ovalbumin. The formation of OVA self-aggregates has been observed after excitation of albumin solutions with 320 nm light (λ_max_ = 400 nm). The results of these experiments show that the light emission of protein aggregates depend linearly on the concentration of OVA (see Appendix A in Appendix A). We obtained the same result for HSA [4] (Figure 1 of Ref. [4]).

Steady-state radiolysis was applied to study the influence of the type of oxidizing radicals on the aggregation process of OVA. The emission spectra of the aqueous solution containing 116 μM OVA in the quartz cell were recorded after irradiation with dose 2200 Gy and are presented in Appendix A. The excitation of irradiated aqueous OVA solution (in the absence and presence of NaN_3_) with the 320 nm light leads to the emission with maximum at 400 nm ($N_{3}^{•}$) and 420 nm (^•^OH), as shown in Appendix A. The maximum of emission of OVA aggregates generated in the irradiated solution by $N_{3}^{•}$ radicals was at the same spectral range as in the case of self-aggregatesemission (before irradiation). The overlapping of the emission bands of both DT and self-aggregates makes it very difficult to identify the formation of dityrosine in OVA solution after irradiation in optical measurements. The excitation of both solutions (irradiated in the presence and absence of NaN_3_) with a wavelength above 320 nm results in emissions above 400 nm. Based on Appendix A, at first glance, the OVA spectra after irradiation under different conditions do not differ. More detailed analysis showed that the mechanism of the azide and hydroxyl radicals’ reactions with OVA differs significantly.

In the case of $N_{3}^{•}$, the wide band is due to the overlapping of the emission of the DT and OVA aggregates (λ_exc_ = 320 nm). We are aware that the reaction of two TyrO^•^ radicals generated in different OVA molecules does not always lead to the formation of fluorescent tyrosine dimer. Regardless of this, in order to achieve a correct identification of DT and aggregates generated by ionizing radiation, it is recommended to analyze the differential spectra obtained by subtracting the OVA emission spectrum recorded before and after irradiation (especially for concentrated albumin solutions) (Figure 9).

When OVA reacts with hydroxyl radical, a similar broad emission band was observed as in the case of the reaction of the azide radical with OVA, but the maximum of this band is red shifted (near 420 nm). As mentioned above, the correction of emission spectra is required. Subtracting the OVA emission spectrum recorded before and after irradiation allowed to distinguish the emission of albumin aggregates. In the case of ^•^OH radicals, species other than DT are formed (maximum of emission at near 420 nm).

The emission spectra recorded for the excitation wavelength 320 nm before and after irradiation of the papain solutions are presented in Figure 10. The emission spectra of irradiated solutions differ slightly depending on the type of reactive species used. As a result of the reaction of azide radicals with papain, tyrosine residues are mainly modified and DT is formed in contrast to ^•^OH radicals, where other species are formed (λ_max_ = 420 nm). The emission spectrum of spontaneous papain dimers does not differ significantly (in contrast to HSA solution) from the emission spectra of aggregates obtained after the reaction of azide radicals with papain. The spectroscopic method allows to distinguish spontaneous dimers to a very limited extent from those generated in the reaction of ^•^OH radicals with papain. In particular, the analysis of emission spectra related to DT formation (attack of azide radicals on papain) is virtually impossible due to the similarity of the emission band of papain self-aggregates to the DT band. It is worth emphasizing that the application of the emission detection method is useful for the observation of the participation of ^•^OH radicals in the reaction with papain. The redshift of the band maximum recorded after irradiation of the papain solution saturated with N_2_O (λ_max_ = 420 nm) indicates that formed aggregates are not stabilized only by DT.

We assume that papain dimers are mainly responsible for the new emission, in the cases of both ^•^OH and $N_{3}^{•}$ radicals. The literature data indicate that papain exists in aqueous solution as a mixture of monomer and dimer, the relative amounts being a function of protein concentration. In addition, electrophoretic and sedimentation studies indicate that papain also has a tendency to aggregate in dimers after oxidation [40]. Our DLS measurements confirm this thesis. Analysis of the dynamic light scattering measurement (DLS) after irradiation of N_2_O-saturated solution containing papain (70 μM) or solution containing papain (70 μM) and NaN_3_ (0.1M) did not confirm the formation of high-molecular aggregates. In the irradiated solutions of papain under oxidizing conditions, papain monomers and dimers are mainly present (d~5 nm). Our measurements are in agreement with the literature [41].

Resonance light scattering (RLS) is a sensitive method for studying protein aggregation. The DLS results showed that the largest aggregates were generated as a result of the attack of the ^•^OH radical on β-lactoglobulin (diameters of several tens of nm), while in the remaining solutions, the size of the nanostructures was below 10 nm. The results of the RLS measurements are in excellent agreement with those obtained by the DLS method. The data in Appendix A show a spectacular 50-fold increase in the RLS signal, resulting from the presence of large aggregates generated as a result of the reaction of ^•^OH radicals with β-lactoglobulin molecules.

The histone solutions (containing or not containing 0.1 M NaN_3_) saturated with N_2_O and irradiated with a dose of 1650 Gy were analyzed spectroscopically by measuring the emission and excitation emission spectra. The emission spectra recorded before and after irradiation of the histone solutions are presented in Figure 11. As a result of the reaction of $N_{3}^{•}$ radicals with histone, tyrosine residues are mainly modified, and DT is formed.

Our measurements are consistent with those described in the literature [42,43]. It was reported that ^•^OH or $N_{3}^{•}$ radicals generated by short electron beam pulses with the reaction of histone were causing intramolecular crossing. In the case of ^•^OH radicals, essentially non-tyrosine moieties of protein were involved. When $N_{3}^{•}$ radicals react with histone during the pulse radiolysis coupling of tyrosine, radicals occur mainly within a single protein molecule. In the case of ^•^OH radical attack on histone, a new emission was observed with a maximum near 400 nm after excitation with 320 nm light of the irradiated solution. The intensity of this band is much lower (5.5 times) compared to the analogous intensity of the spectrum recorded after the reaction of $N_{3}^{•}$ radicals with protein. It should be noted that the excitation of saturated N_2_O and irradiated histone solution with light at above 320 nm leads to emission in the spectral range of 430–530 nm. In the case of $N_{3}^{•}$ radicals, the new emission in this spectral range is negligible. These results suggest that the reaction of ^•^OH radicals with histone leads to the formation of both intra- and intermolecular bonds between histone molecules.

The use of a more accurate measurement technique based on light scattering should help distinguish intra- and intermolecular bonds between tyrosine residues. Analysis of the light scattering intensity (LSI) after irradiation of N_2_O-saturated solution containing histone indicate intramolecular cross-linking of tyrosine residue [44]. It is notable that intermolecular cross-linking mainly involves non-tyrosine histone moieties. The azide radicals react rather selectively with several histone residues, especially with tyrosine and cysteine moieties. When azide radicals react with the histone, the LSI increased weakly and the formation of dimers was strongly impeded.

### 3.3. The Formation of Dityrosine in Peptide Solution by Ionizing Radiation

Recent studies have provided the first indication that HSA can be used as a drug delivery system for peptide analogs of insulin hot spots, inhibiting insulin aggregation [36]. It was shown that the binding of VEALYL (short peptide with amino acids sequence: Ala-Glu-Ala-Leu-Tyr-Leu) by HSA prevents the formation of amyloid fibrils. In this work, we conducted studies on the influence of oxidative stress on properties of VEALYL, which may help to understand the aggregation process of more complex systems as proteins and enzymes. It is easy to predict that the key amino acid involved in scavenging oxidative radicals in the case of VEALYL is tyrosine. Reactions of $N_{3}^{•}$ or ^•^OH radicals with tyrosine over a wide range of pH and amino acid concentrations are well-known and we recently reported the detection of DT formation by steady-state and time-resolved fluorescence measurements [4].

To study the reaction of an ^•^OH radical with VEALYL, we applied pulse radiolysis. Transient absorption spectra recorded at two selected times after an electron pulse irradiation of N_2_O-saturated aqueous solution containing 70 μM VEALYL is presented in Figure 12. The broad absorption band with maximum near 330 nm was observed immediately after irradiation (black plot). This spectrum can be attributed to the absorption of the tyrosine OH adduct on the ortho-position to the OH group. The band with a maximum at~330 nm disappears and a new transient absorption peak with maximum at 410 nm is formed (red plot). Pulse radiolysis confirmed that the reaction of a hydroxyl radical with peptide resulted in the formation of the phenoxyl radicals TyrO^•^, which absorbs near 410 nm. A similar conclusion regarding the formation of DT in N_2_O-saturated VEALYL solution was drawn in the case of the irradiation of H-Gly-Tyr-Gly-OH (GYG) tripeptide oxidized with ^•^OH radicals [12]. Authors reported that the main route of phenoxyl radical formation consists of the addition reaction of hydroxyl radical to the phenol ring on the tyrosine side-chain and proton-catalyzed water molecule elimination.

When insulin reacts with hydroxyl radicals, similar transient absorption spectra were obtained as in the case of the reaction of ^•^OH with VEALYL. For insulin, the absorption band associated with the maximum at 330 and 410 nm is less intense as for the VEALYL (spectra not shown) [39].

The comparison of the absorption spectra recorded during the pulse radiolysis of aqueous solutions of VEALYL or insulin containing 0.1 M NaN_3_ indicate that the mechanism of oxidation of these compounds by azide radicals is similar (Figure 13). A significant difference was observed in the 350–450 nm region. The higher absorbance in the transition spectra for VEALYL after one-electron oxidation resulted from the accessibility of tyrosine residues for $N_{3}^{•}$ radicals. When the oxidizing radical was ^•^OH, we observed the formation of various types of tyrosine radicals (OH adduct on the ortho-position to the OH group and TyrO^•^), and the oxidation of VEALYL and insulin with the azide radical led mainly to the formation of TyrO^•^.

Emission spectra of N_2_O-saturated VEALYL peptide solution (470 µM) and insulin solution (140 µM) after electron beam irradiation (1700 Gy) are presented in Appendix A (recorded after excitation with 320 nm light). An important observation from our steady-state radiolysis measurements was the presence of DT in the irradiated VEALYL peptide solution [4]. The analysis of the measurement data showed that the emission intensity observed in the spectral range of 400–410 nm from DT depends on the spatial accessibility of tyrosine residues within insulin and the VEALYL peptide by ^•^OH radicals generated in the solution.

The structure of the VEALYL peptide contains one Tyr residue, which is exposed to an aqueous solution and reacts easily with radicals, which explains the difference in the intensity of the DT band compared to the insulin solution (14 times higher peptide emission intensity compared to insulin). In a solution, insulin has the unique trait of assuming different association states, including dimers, tetramers and hexamers [45]. Insulin is composed of two chains, an A chain (with 21 amino acids) and a B chain (with 30 amino acids), which are linked by two disulfide bridges. Insulin contains four tyrosine residues, two of which are in the A peptide chain and other two in the B chain. In the monomer of insulin, there are four surface-exposed residues of Tyr. After self-aggregation, two Tyr residues are partially buried (Tyr16, B chain) and two residues are buried deeply within the protein structure (Tyr26, B chain). For this reason, in spectroscopic measurements, differences were observed in the intensity of DT emission between insulin and VEALYL peptide.

Appendix A (see in the Appendix A) shows the normalized emission spectra of peptide and insulin. We confirm that peptide and insulin aggregate under oxidative stress. In both cases, covalent bonds are formed between the tyrosine residues. Insulin aggregates which form as a result of the reaction of albumin with ^•^OH radicals are stabilized by hydrogen bonds. The emission spectrum observed in this case is wider than the classic DT band. This is due to the overlap of dityrosine fluorescence and the emission of aggregates that are formed by a network of hydrogen and/or ionic bonds, analogous to self-aggregates.

## 4. Discussion

Recently, using pulse radiolysis [4], we compared the reactions of oxidative radicals with albumin, and these radicals with selected components of HSA (e.g., tryptophan, tyrosine, phenylalanine, cystine, histidine). We recorded transient absorbance spectra for all studied objects. We are able to attribute these absorption bands to the products formed as a result of the oxidation of Tyr and Trp in albumin by azide radicals: the band of the Tyr meta isomer centered at 305 nm and the TyrO^•^ phenoxy radicals with a maximum of about 410 nm. The band with a maximum at around 520 nm is attributed to the oxidation of the tryptophan molecule, Trp^•^. The band with the maximum at 275 nm can be related to the reaction product of the azide radical with phenylalanine [4] (Figure 22 of Ref. [4]).

In this work, for all studied objects (papain, OVA, insulin, β-lactoglobulin, lysozyme, histone), pulse radiolysis measurements were performed and transient absorbance spectra were recorded for their reaction with the azide radical. Since the same amino acids are components of the proteins and enzymes studied here, as a result of radiolysis, similar transient spectra to those recorded for albumins were to be expected. The analysis of the obtained transient absorbance spectra allowed us to establish that this is the case. On the other hand, the intensities of the individual primary bands of the spectrum of the $N_{3}^{•}$ reaction product with our object differ from those previously recorded for albumins. This is due to two main reasons. The first of them results from the different content of amino acids that play a leading role in the transient absorbance spectra. These amino acids are important aromatic amino acids (tryptophan, tyrosine, phenylalanine) due to their high reactivity with the azide radical, and additionally, due to the high molar values of absorbance coefficients of their oxidation products. The second reason is related to the structure of proteins and enzymes, namely the access of individual amino acids for radicals penetrating protein objects from the aqueous phase. In our opinion, pulse radiolysis measurements are crucial for understanding the mechanism of protein oxidation. These studies allow us to determine not only which amino acids are involved in the process of oxidative radical attack, but also what is the sequence of electron transfer. Measurements of absorption spectra and, above all, the emission spectra of irradiated solutions allow to determine the final sites of protein damage. Recently, we have shown that in the reaction of $N_{3}^{•}$ radicals with BSA, the intensity of the absorption band originating from Trp residues is twice as intense as that recorded for the HSA solution. This indicates that both tryptophan residues in BSA are accessible to azide radicals. In the case of an OVA reaction, a larger amount of Trp^•^ radicals is also generated than for an HSA oxidation reaction. This is probably associated with a larger number of tryptophan residues in OVA (three Trp residues ) in relation to HSA (only one Trp unit). After normalizing the spectra obtained in the initial stage of the oxidation of the examined objects (Figure 3), three main bands (with maxima 520, 410 and 305 nm) are observed, but the spectra are not identical. The correlation of the shape of the spectrum of the primary oxidized samples with the ratio of the number of tryptophan and tyrosine residues is clearly observed. The intensity of the band (spectrum normalized to the radical band of the tryptophan residue) associated with the Tyr^•^ radical is the lowest for lysozyme (smallest tyrosine residue/tryptophan residue ratio: 0.5 (3/6)). For papain and OVA, the pulse radiolysis transient spectra are similar and the Tyr/Trp ratio values are similar: 3.8 for papain (19/5) and 3.33 (10/3) for OVA. The spectrum of the reaction of $N_{3}^{•}$ with albumin is similar in shape to that of OVA and papain, but the bands for Trp^•^ or TyrO^•^ residues are more clearly exposed than the spectrum of other oxidation products of other protein amino acids. The pulse radiolysis spectra presented in Figure 3 were presented in such a way that the bands from the two main products of the $N_{3}^{•}$ reaction, i.e., TyrO^•^ and Trp^•^, were sufficiently intense for analysis. In the case of papain oxidation, the experiment for high radiation dose is shown. The measurement made for a concentration of 135 µM and a dose of 55 Gy allows the identification of two leading primary products (TyrO^•^ and Trp^•^), but the spectrum is noise and of low intensity (see Appendix A in Appendix A). It is worth emphasizing that the time of recording a given transient absorbance spectrum is important, because the $N_{3}^{•}$ reaction rate constants with tyrosine or tryptophan residues in the protein may differ in value. Fortunately, the reaction rate constants of the $N_{3}^{•}$ radical with the tested proteins have similar values and are close to 6·10^8^ dm^3^ mol^−1^ s^−1^ (for example, 6·10^8^ dm^3^ mol^−1^ s^−1^ for β-lactoglobulin (Trp), 9.5·10^8^ dm^3^ mol^−1^ s^−1^ for HSA(Trp), 6.5·10^8^ dm^3^ mol^−1^ s^−1^ for papain (Trp), 1·10^9^ dm^3^ mol^−1^ s^−1^ for HSA(Tyr)). These data are in good agreement with [16,17].

A very small share of the tyrosine radical spectrum in the radiolysis spectrum of the lysozyme solution with the $N_{3}^{•}$ radical may suggest that protein aggregates induced by the Tyr–Tyr bridge will not be formed. Our emission measurements (Figure 7) clearly contradict this, and the emission band associated with DT is very intense. An apparent paradox—the absence of a TyrO^•^ radical in pulse radiolysis measurements at the final Tyr–Tyr bridge formation—can be easily explained. The tyrosine radical is generated in a longer time scale (ms) as a result of the charge transfer from the tyrosine moiety to Trp^•^ radical, as shown in Figure 1. In the analysis of the formation of DT (Tyr–Tyr) bridges, the charge transfer process involving the Trp^•^ radical should be taken into account. Pulse radiolysis results provide helpful information on inter- and intramolecular electron transfer in protein solutions. It is worth emphasizing that the inclusion of the product analysis method in addition to radiation techniques may allow us to determine which amino acids are involved in electron transport. An example of a more detailed insight into the process of radical transformation Trp^•^ → TyrO^•^ for various lysozymes can be found in [14]. The authors observed electron transfer within hen egg-white lysozyme between the following amino acid residues: Trp62/Tyr53, Trp63/Tyr53 and Trp123/Tyr23. On the other hand, tryptic hydrolysis, followed by HPLC and MALDI-TOF MS measurements, performed on a lysozyme solution containing NaN_3_ (0.1 M) and γ- irradiated indicate that Trp108 and/or Trp111 remain oxidized, and that Tyr20 and Tyr53 result in DT. There is evidence that some tryptophane residues and some tyrosine residues are not involved in the LRET process within the protein structure. Generally, for all six proteins, the LRET is not stoichiometric and other residues are involved in charge transfer, which is silent in pulse radiolysis. Such intramolecular electron transfer is effective in many systems [16], for example, in β-lactoglobulins (see Appendix A in Appendix A), but in the case of albumins (according to the literature [22]), it is ineffective and slow, taking place in the seconds. An important conclusion from the analysis of the formation of protein aggregates under the influence of ionizing radiation is the tracking of the oxidation process in a very long time scale.

Resonance light scattering (RLS) is an extremely sensitive and useful technique to detect aggregates. RLS spectra can be recorded by scanning both excitation and emission monochromators of a typical spectrofluorometer with Δλ = 0. The six solutions used in the DLS measurements were also used in the RTL measurements. The DLS results showed that the largest aggregates were generated as a result of the attack of the ^•^OH radical on β-lactoglobulin (diameters of several tens of nm), while in the remaining solutions, the size of the nanostructures was below 10 nm.

The reaction rate constants of the $N_{3}^{•}$ radical with the tested proteins have similar values and are close to 10^9^ dm^3^ mol^−1^ s^−1^. For β-lactoglobulin, this reaction rate constant is equal to 6.5 × 10^8^ dm^3^ mol^−1^ s^−1^ for HSA 10^9^ dm^3^ mol^−1^ s^−1^.

In our recent paper, we report that the hydrogen bond and/or π–π stacking interactions are the cause of intrinsic blue autofluorescence in protein aggregates in solutions. In addition, we have shown that two types of self-aggregates are formed. The existence of two groups of aggregates was confirmed in the measurements of light emission, steady-state and time-resolved. For proteins and enzymes with a lower molecular weight than HSA, the presence of one group of aggregates dominates, namely those associated with the short-wave band of emission spectra. It should be emphasized that the nature of self-aggregates for small proteins, enzymes or polypeptides is the same as for albumins, only both groups of aggregates are noted for higher concentrations in relation to albumins. This thesis is confirmed by the recorded emission and excitation spectra for a highly concentrated neat β-lactoglobulin solution (see Appendix A in Appendix A).

## 5. Conclusions

In our opinion, the detection of luminescence resulting from the UV and Vis light excitation of protein aggregates is useful in their characterization. All the objects we study show emission caused by spontaneous aggregation. Aggregates are formed even for low protein concentrations, without the need to exceed the critical concentration. In this work, we confirmed, in line with our previous observations made for HSA, that the emission intensity of spontaneous aggregates increases monotonically with the concentration of proteins in solution. The occurrence of this emission becomes an obstacle in the analysis of newly formed luminescence resulting from the excitation of nanostructures generated by the use of ionizing radiation. Regardless of this, in order to correct the identification of DT and aggregates generated by ionizing radiation, it is recommended to analyze the differential spectra obtained by subtracting the emission spectrum recorded before and after irradiation. A thorough analysis of the emission data for studied proteins and enzymes allows to distinguish whether the cause of luminescence is the presence of aggregates initiated by the reaction with the ^•^OH or $N_{3}^{•}$ radical. A good example of this is the analysis of the formation of aggregates induced by the reaction of the ^•^OH or $N_{3}^{•}$ radical with OVA. As shown in Appendix A, the emission spectra of OVA solutions irradiated under conditions where the ^•^OH or $N_{3}^{•}$ radical is reactive are very similar. The method of the analysis of emission spectra proposed by us showed that the reaction mechanism of azide and hydroxyl radicals with OVA differs significantly. We can unequivocally conclude that the reaction of OVA with $N_{3}^{•}$ leads to the formation of cross-linked DT, while in the case of the reaction of OVA with ^•^OH, other aggregates dominate (Figure 9). This procedure is useful for analyzing the effect of ionizing radiation on other studied systems. In the case of lactoglobulin, histone or papain, the analysis of the newly formed emission of aggregates as a result of irradiation is more difficult and requires the precise correction of the share of primary emission (the one observed before irradiation) because it is centered around 400 nm (band typical for DT). In our opinion, the use of time-resolved techniques to determine the emission lifetimes of aggregates makes it possible to distinguish protein aggregates obtained as a result of irradiation from spontaneous aggregates.

We have shown that pulse radiolysis measurements are crucial for understanding the mechanism of protein oxidation. These experiments confirmed that the mechanism of the reaction of azide radicals with proteins and enzymes is significantly different from the processes associated with the attack of hydroxyl radicals on this protein. We found that as a result of the reaction of $N_{3}^{•}$ with the studied proteins, Tyr–Tyr bridges (DT) are formed, which leads to the formation of mainly protein dimers. High-molecular aggregates stabilized by hydrogen bonds or ionic bonds are observed in the case of the ^•^OH radical. The high reactivity of ^•^OH with amino acids in proteins explains the possibility of the formation of covalent bonds other than Tyr–Tyr bridges (including C–C or C–O–C) between adjacent protein molecules. The exception turned out to be papain, which forms mainly dimers after irradiation (both after reaction with ^•^OH and $N_{3}^{•}$ radicals). The results of the pulse radiolysis measurements were confirmed in dynamic light scattering measurements. The results of our RLS measurements are in good agreement with those obtained by DLS technique and show that large aggregates are formed in the case of studied proteins after reaction with ^•^OH radicals.

Intramolecular electron transfer from the tyrosine residue to the Trp^•^ radical is very important in the process of the formation of aggregates under the influence of ionizing radiation. An excellent example of this are the pulse radiolysis measurements made for a lysozyme solution. Our steady-state emission measurements show that the reaction of the azide radical with lysozyme leads to dimer formation via DT. The analysis of pulse radiolysis measurements allowed to conclude that, initially, the Trp^•^ radical is generated, and then, as a result of LRET, the TyrO^•^ radical is formed. Recombination of this radical with a twin radical from another oxidized lysozyme molecule results in the formation of a dimer.

## Figures and Tables

**Figure 1 pharmaceutics-15-01367-f001:**
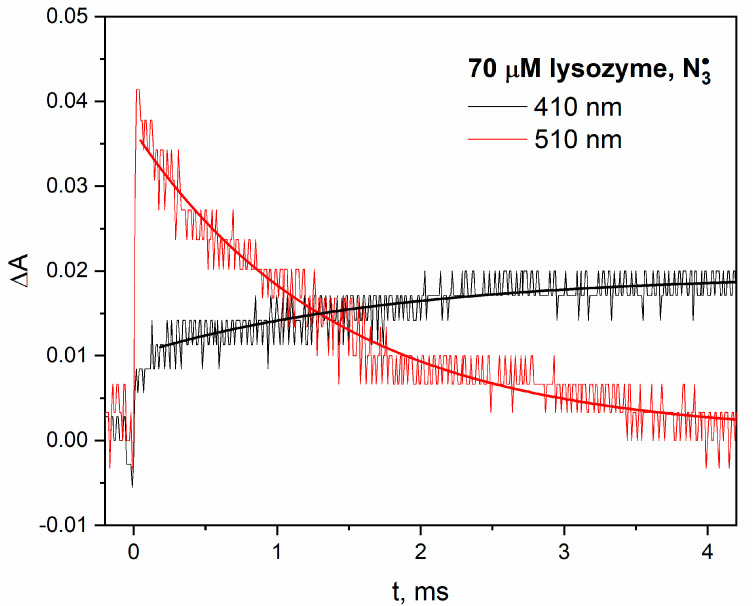
Time profiles of the absorbance recorded at 410 or 510 nm after irradiation with a dose of 200 Gy of the N_2_O-saturated aqueous solution containing lysozyme (70 µM) and NaN_3_ (0.1 M). The kinetics of these bands is described by single exponential decay.

**Figure 2 pharmaceutics-15-01367-f002:**
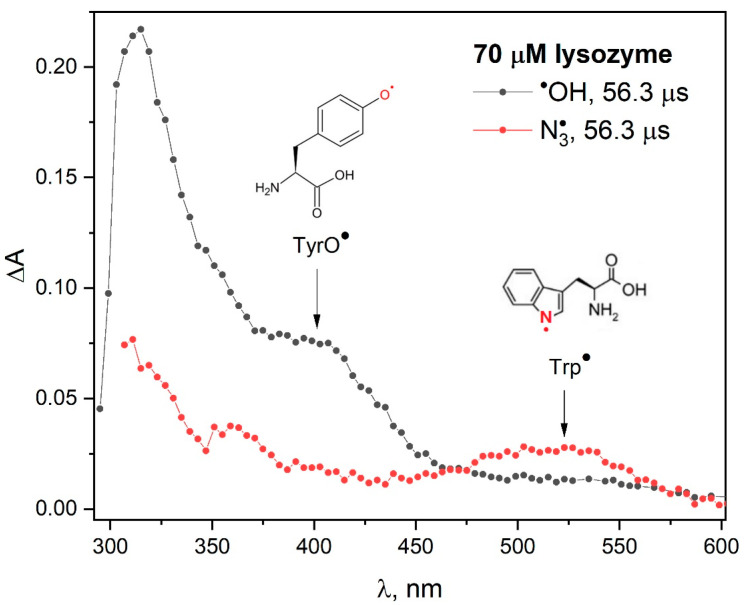
Transient absorption spectra of N_2_O-saturated buffer solutions containing: 70 µM lysozyme and 0.1 M NaN_3_, obtained for irradiation dose of 240 Gy.

**Figure 3 pharmaceutics-15-01367-f003:**
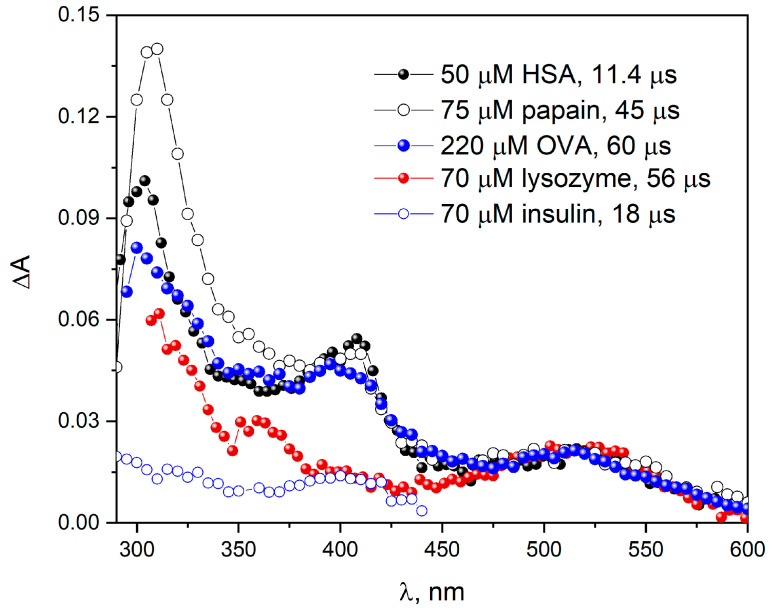
Transient absorption spectra of N_2_O-saturated buffer solutions containing 0.1 M NaN_3_ and: HSA (50 µM); OVA (226 μM), lysozyme (70 µM), insulin (70 µM); obtained for irradiation dose of 200 Gy. Transient absorption spectra of N_2_O-saturated buffer solutions containing 0.1 M NaN_3_ and papain (75 µM), obtained for irradiation dose of 800 Gy. For the sake of comparison, the spectra are normalized based on the intensity of the absorbance band at 510 nm.

**Figure 4 pharmaceutics-15-01367-f004:**
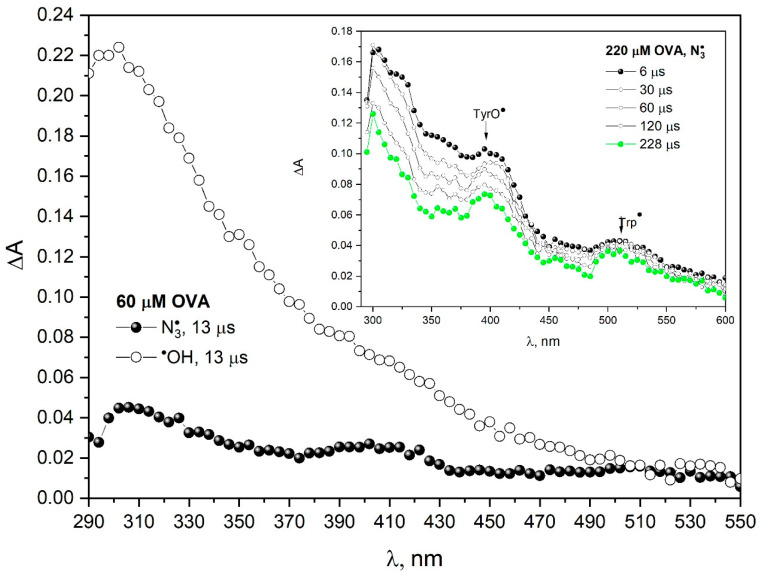
Transient absorption spectra of N_2_O-saturated buffer solutions containing 60 μM OVA and 0.1 M NaN_3_, obtained for irradiation dose of 55 Gy (black circle). Transient absorption spectra of N_2_O-saturated buffer solutions containing 60 μM OVA, obtained for irradiation dose of 55 Gy (white circle). (Insert). Transient absorption spectra of N_2_O-saturated buffer solutions containing 220 μM OVA and 0.1 M NaN_3_, obtained for irradiation dose of 200 Gy.

**Figure 5 pharmaceutics-15-01367-f005:**
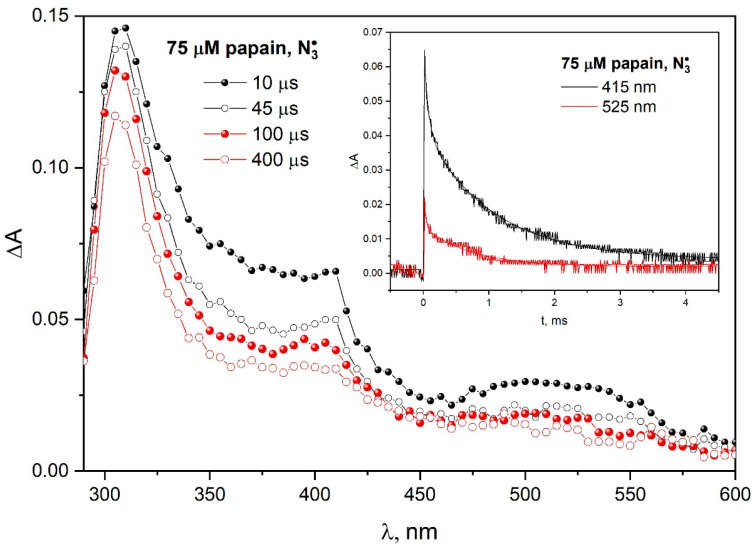
Transient absorption spectra of N_2_O-saturated buffer solution containing 75 µM papain and 0.1 M NaN_3_, obtained for an irradiation dose of 800 Gy. (Insert). Time profiles of the absorbance recorded at 415 or 525 nm after irradiation with a dose of 800 Gy of the N_2_O-saturated aqueous solution containing papain (75 μM) and NaN_3_ (0.1 M).

**Figure 6 pharmaceutics-15-01367-f006:**
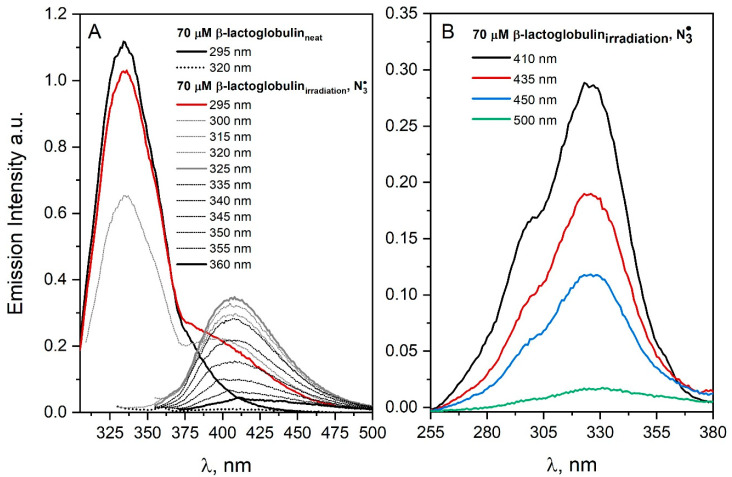
(**A**) Emission spectra of the N_2_O-saturated β-lactoglobulin solution (70 µM) containing 0.1 M NaN_3_, recorded before and after irradiation with a dose 250 Gy. The excitation wavelengths are given in the figure. (**B**) Emission excitation spectra of the N_2_O-saturated β-lactoglobulin solution (70 µM) containing NaN_3_ (0.1 M), recorded after irradiation with a dose 250 Gy. The emission wavelengths are given in the figure.

**Figure 7 pharmaceutics-15-01367-f007:**
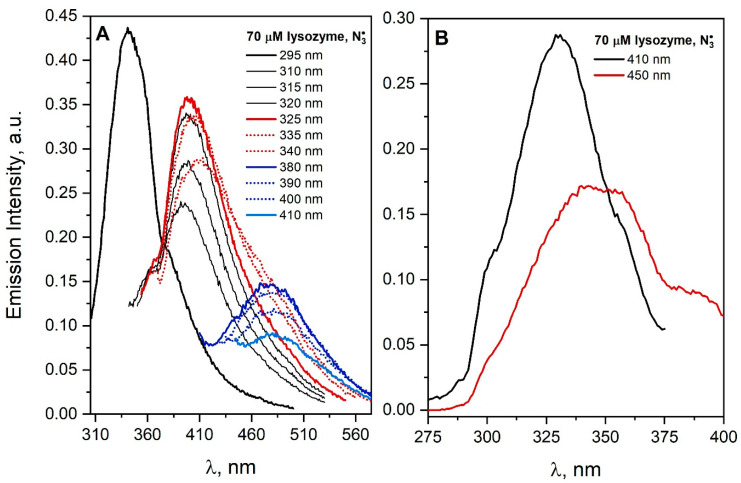
(**A**) Emission spectra of the N_2_O-saturated lysozyme solution (70 µM) containing NaN_3_ (0.1 M), recorded after irradiation with a dose 3200 Gy. The excitation wavelengths are given in the figure. (**B**) Emission excitation spectra of the N_2_O-saturated lysozyme solution (70 µM) containing NaN_3_ (0.1 M), recorded after irradiation with a dose 3200 Gy. The emission wavelengths are given in the figure. The width of the slits provided a spectral resolution of 8/2 nm.

**Figure 8 pharmaceutics-15-01367-f008:**
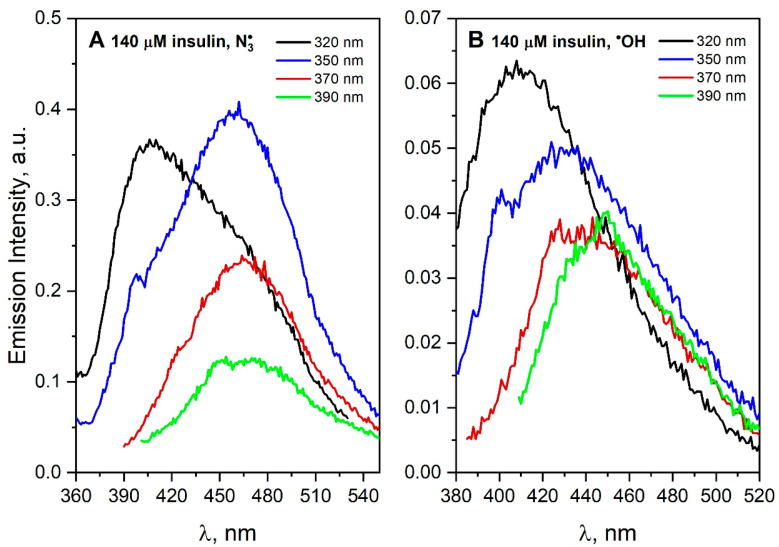
(**A**) Emission spectra of N_2_O-saturated buffer solution of insulin (140 μM) containing NaN_3_ (0.1 M) recorded after irradiation with dose 1485 Gy. (**B**) Emission spectra of N_2_O-saturated buffer solution of insulin (140 μM) recorded after irradiation with dose 1485 Gy. The excitation wavelengths were 320, 350, 370 and 390 nm.

**Figure 9 pharmaceutics-15-01367-f009:**
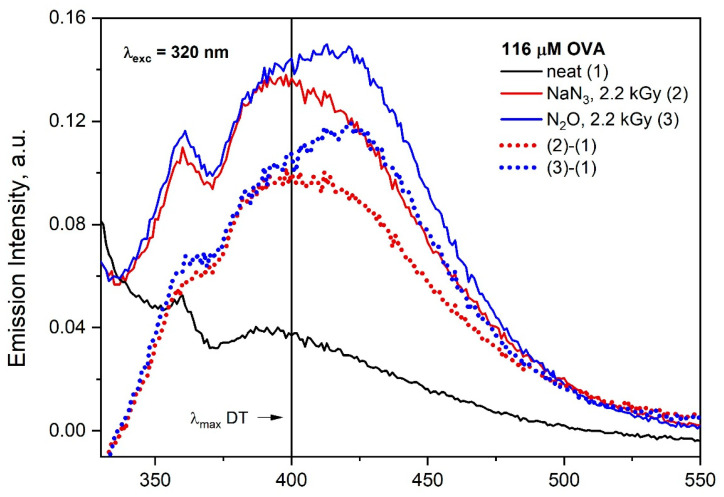
Emission spectra of N_2_O-saturated buffer solutions of OVA (116 μM) in the absence and presence of NaN_3_ (0.1 M) recorded after irradiation with dose 2200 Gy. The excitation wavelength was 320 nm.

**Figure 10 pharmaceutics-15-01367-f010:**
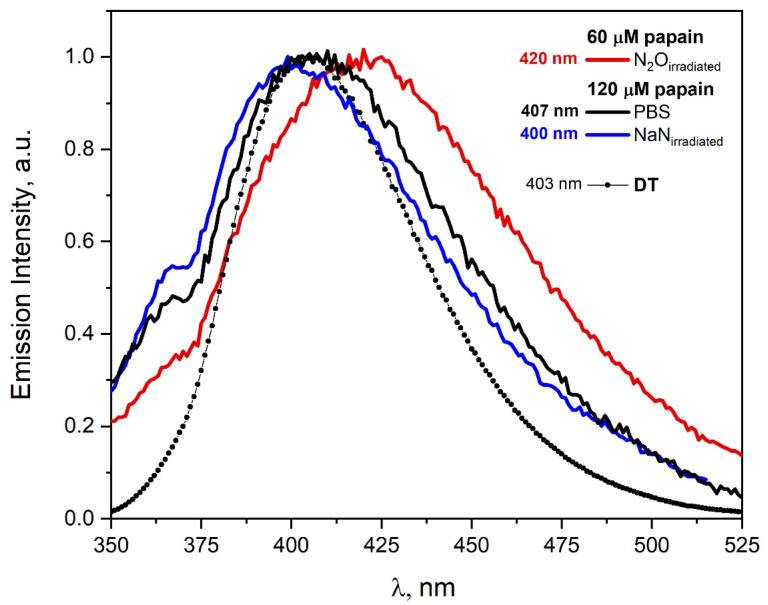
Emission spectra of N_2_O-saturated buffer solution of papain (60 μM) recorded after irradiation with dose 3120 Gy. Emission spectra of N_2_O-saturated buffer solution of papain (120 μM) containing NaN_3_ (0.1 M) recorded before and after irradiation with dose 3120 Gy. Emission spectra of N_2_O-saturated Tyr solutions (2 mM) containing NaN_3_ (0.1 M) recorded after irradiation with a dose 60 Gy. The excitation wavelength was 320 nm. Spectra were normalized (I = 1.0).

**Figure 11 pharmaceutics-15-01367-f011:**
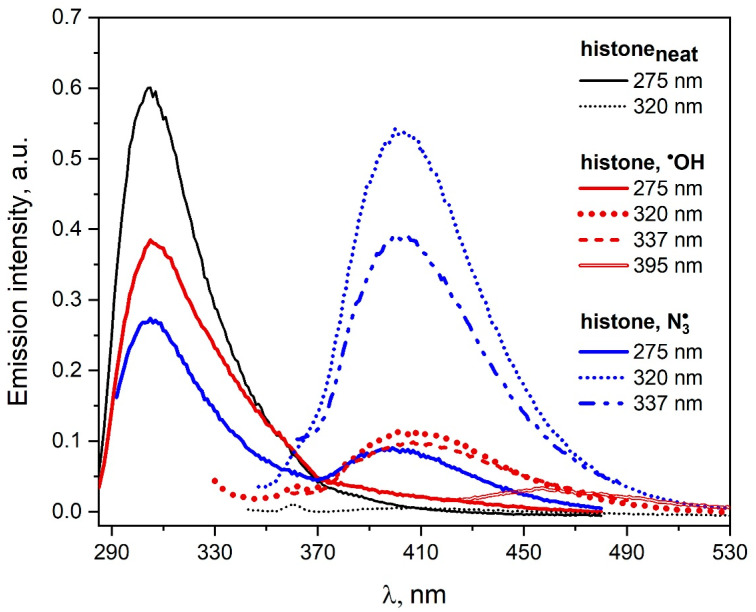
Emission spectra of the neat histone solution (1.4 mg/mL, pH = 7.2) and N_2_O-saturated histone solutions (1.4 mg/mL) containing NaN_3_ (0.1 M) recorded before (black lines) and after irradiation with a dose 1650 Gy (in the case of $N_{3}^{•}$—blue lines, in the case of ^•^OH—red lines). The excitation wavelengths are given in the figure.

**Figure 12 pharmaceutics-15-01367-f012:**
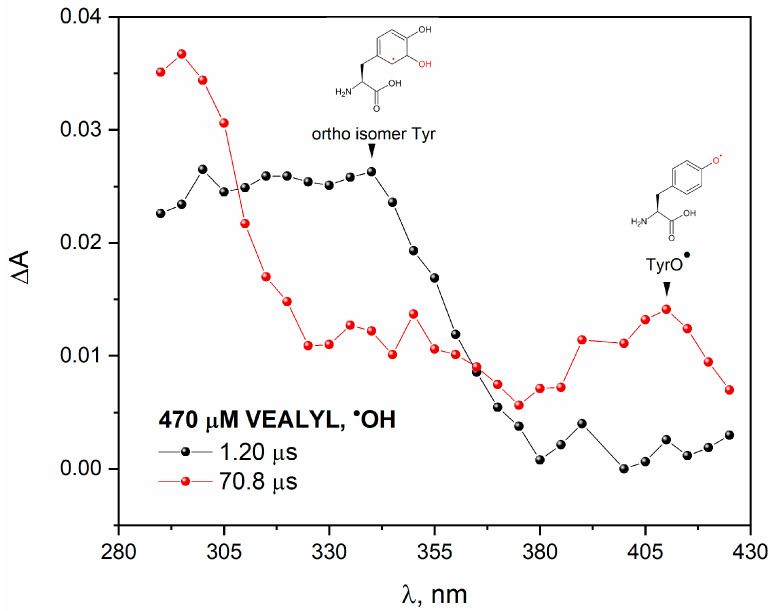
Transient absorption spectra of N_2_O-saturated buffer solutions containing 470 μM VEALYL obtained for an irradiation dose of 55 Gy.

**Figure 13 pharmaceutics-15-01367-f013:**
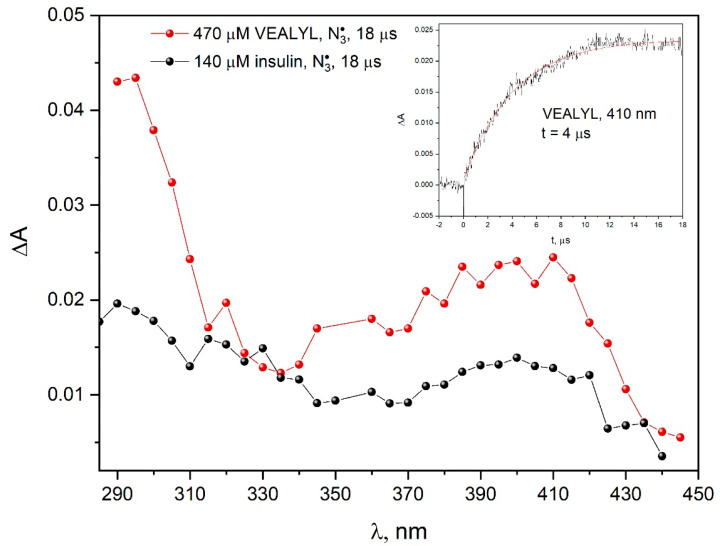
Transient absorption spectra of N_2_O-saturated buffer solutions containing 470 μM VEALYL and 0.1 M NaN_3_ obtained for an irradiation dose of 55 Gy (red plot). Transient absorption spectra of N_2_O-saturated buffer solutions containing 140 μM insulin and 0.1 M NaN_3_ obtained for an irradiation dose of 55 Gy (black plot). (Insert): Time profiles of the absorbance recorded at 410 nm after 17 ns pulse irradiation with a dose of 55 Gy of the N_2_O-saturated aqueous solution containing VEALYL (470 µM) and NaN_3_ (0.1 M).

## Data Availability

Not applicable.

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
