# Peer review of "Influence of Ionizing Radiation on Spontaneously Formed Aggregates in Proteins or Enzymes Solutions"

_pharmaceutics, 2023, doi:10.3390/pharmaceutics15051367_

Round 1
Reviewer 1 Report
Please see the attached document.

Author Response
Please find attached the responses to the Reviewer's comments. Text written in bold type we proposed to be included in the manuscript.

Reviewer 2 Report
I reviewed the manuscript entitled: “Aggregation-induced emission in neat and irradiated solutions of proteins and enzymes: detection of protein nanoparticles” (Manuscript ID: pharmaceutics-2295408) submitted to Pharmaceutics. This paper describes the emission of dimerized proteins by oxidation through gamma ray irradiation. Radical reactions are examined and discussed and the reaction rate constants are presented. Noticed points are listed as follows:
1) For discuss the protein emission, the chromophore responsible for these emissions are very important. The identification of these chromophores must be characterized. Chemical structure of the chromophores should be explained.
2) Lines around 172: The electron transfer is unclear. The electron donor and acceptor should be mentioned clearly.
3) Title: “Aggregation-induced...” is appropriate? If the aggregation induces the chromophore formation of these emission, this title is appropriate. If the dimerization through the cross-linking is the mechanism of the chromophore formation, other tittle may be more appropriate.
4) Results section: The style of this section is “Results and discussion”. The sections should be updated. For example, the “Results” section should be updated to “Results and discussion”, the “Discussion” should be separated to “Mechanism of chromophore formation” in the Results and discussion section and “Conclusion” section.
5) (Minor point) Lines around 89: Absorption coefficients are provided. The wavelengths should be presented for these absorption coefficients.
6) (Minor point) Line 140: The abbreviation “DT” should be opened in the main text not only abstract.
Author Response

(The authors gave the same response as above.)

Round 2
Reviewer 2 Report
I confirmed the revised form.
Author Response
Thank you for your valuable comments